# Radio Communications on Family Planning: Case of West Africa

**DOI:** 10.3390/ijerph19084577

**Published:** 2022-04-11

**Authors:** Jaehyun Ahn, Gary Briers, Mathew Baker, Edwin Price, Robert Strong, Manuel Piña, Alexis Zickafoose, Peng Lu

**Affiliations:** 1Department of Agricultural Leadership, Education, and Communications, Texas A&M University, College Station, TX 77843, USA; mathew.baker@ag.tamu.edu (M.B.); robert.strongjr@ag.tamu.edu (R.S.); manuel.pina@ag.tamu.edu (M.P.); alexis.zickafoose@ag.tamu.edu (A.Z.); peng.lu@ag.tamu.edu (P.L.); 2Department of Agricultural Economics, Texas A&M University, College Station, TX 77843, USA; edwin.price@ag.tamu.edu

**Keywords:** family planning, radio communications, mothers, demographic and health surveys, West Africa, population growth

## Abstract

Sub-Saharan Africa will accommodate more population this century by having a multitude of births across the continent. Family planning methods provide women with techniques to manage their health and wellbeing. This study investigated how radio communications in family planning changed the perception of Ghanaian, Liberian, and Senegalese mothers toward having fewer children. Univariate and multivariate linear regression results after coarsened exact matching (CEM) with selected covariates for 15- to 49-year-old mothers from demographic and health survey (DHS) data implied the effectiveness of radio communications. This effort supports the need for further research on tailored communication methods for West African mothers over time.

## 1. Introduction

The population clock is ticking faster. In two decades, the world has increased in population by a third and is expected to exceed 8 billion in the next decade. The trend will continue to approximately 10.9 billion by 2100 [1]. With further inspection, there remain fundamental differences. Fertility rates in East Asia, Europe, and North America will stall at fewer than 2.1 births per mother by 2050. The median age will rise to more than 40 years. The increase in the age dependency ratio will be the onus for elderly dependence onto others of working age. Africa faces another age asymmetry. Sub-Saharan countries will deliver 1.3 billion new births of the 2 billion globally in three decades. Therefore, a quarter of the global population will be in Africa. The fertility rate of sub-Saharan Africa (SSA) will be more than double the East Asia, Europe, and North America fertility rate [2,3,4].

In 2020, many 15- to 49-year-old women of reproductive age and their partners delayed or avoided pregnancy [5]. More use of modern contraception ensued in the Northern Hemisphere [5]. In the Southern Hemisphere (SSA, Melanesia, Micronesia, and Polynesia), half of all mothers showed no avoidance, postponements of pregnancy, or contraceptive use. Contraceptive knowledge and use tend to be a couple’s decision rather than an individual one. Many mothers in SSA miss learning and implementing those lessons and confront childbearing and caring challenges. More than one of every five women in SSA becomes a mother before 16 years old. Poverty is the root cause of teenage mothers and their children’s educational, medical, and other social inequities [6].

The governments in sub-Saharan Africa try to balance uneven progress. For instance, *Baby by Choice, Not by Chance* was a slogan used for World Contraceptive Day in which Liberian couples heard, watched, or received information on contraception. The Liberian government and the United States Agency for International Development (USAID) guide women to access modern contraceptive methods. *Family Planning is Good for Baby Ma!* educational posters hung in villages and hospitals give couples opportunities to know about family planning. The concerted effort supports the need for family planning in poverty-stricken and food-insecure regions [7,8]. Long-term support should aim for young village mothers to be free from sexual assault and intimate partner violence during pregnancy and childcare [9,10,11]. This study’s target populations were the same as the national campaigns: 15- to 49-year-old women who are married or cohabitating with partners. Simultaneously, they are the ones who make family planning decisions in support of and with their husbands/partners.

Ghana, Liberia, and Senegal were three western SSA countries selected for the study. They are located close together but differ from one another. The population clock ticks differently in that Ghana, Senegal, and Liberia ranked 13th, 27th, and 37th of the top 50 most populous African countries in 2022 [12]. Simultaneous inclusion of these three also represents different religions, literacy, education levels, and more to help reduce study bias and enhances the evidence toward action.

Radios in SSA are non-discriminate, widespread, and effective for villagers and urbanites. Local radio stations equipped with information and communication technologies (ICTs) will deliver well-suited indigenous knowledge with up-to-date information [13]. Women will express their own messages in the process of communication. The central hypothesis resides in the power of radio communications in family planning. The aim of increasing public awareness of family planning is to investigate more realistically and compare the effectiveness of radio on contraception, loss of children, and the desire for family planning. If the results and interpretations suffice, this study could be evidence for nationwide family planning campaigns reinforcing radio communication.

Demographic and health surveys (DHS) from USAID were the study’s primary data that have represented invaluable household information over 90 low- and middle-income countries worldwide since 1984. Despite multiple survey developments, the focus has been steadfast on fertility, mortality, reproductive health, and family planning. High responses to the enhanced and newly added questions for 15- to 49-year-old women in sociodemographic, nutritional, and socioeconomic status make meticulous comparative research possible between and among countries [14].

DHS information motivated this study. Responses in the 2013 Liberia DHS showed many more women between 15 and 49 years old participated or heard *Baby by Choice, Not by Chance* and *Family Planning is Good for Baby Ma!* campaigns after survey-weighting (19,007 yes (71%) vs. 7538 no (28%)). The rest of the 120 respondents (1%) answered that they did not know or left the responses empty. By residence, more urban women accessed the campaigns (10,739/13,211 urban respondents, 82%) than rural women (7664/13,454 rural respondents, 57%). Both rural and urban women (18,403 urban and rural respondents) responded that radio was their main campaign-information resource. All these statistics indicate the effectiveness of radio communications regardless of residence.

Additional information from the 2008 Ghana DHS substantiates the need for family planning. As Ghanaian females represent autonomous individuals in health, education, and economic activities, 1016 out of 1216 urbanites (86%) and 1315 of 1660 villagers (79%) answered *having too many children is dangerous for women*. Leaning on consecutive statements (1) *better not to have more children than can be afforded* (2) *children from smaller families are more likely to succeed* affirms the role of women in family planning (1138/1216 urbanites (94%) and 1503/1660 villagers (91%) agreed on (1); 1028/1216 urbanites (85%) and 1347/1660 villagers (81%) believed (2) as a true statement).

In another question on the role of contraception, about 63% (1809/2877 rural and urban respondents) answered contraception is the responsibility of both men and women compared with 33% (946/2877) who responded that contraception is a woman’s responsibility (4% or 122 respondents preferred not to answer). Although the descriptive information indicates the need for autonomy of females or mutual decision-making between partners on family planning, the reality reflects the opposite in varied modern contraception in urban and rural areas in Senegal, Liberia, and Ghana. That is the reason more and better public sector intervention is imperative [15,16,17,18,19,20,21,22,23,24,25,26].

This study presents a novel statistical matched multivariate regression to delineate the effectiveness of radio communications in family planning. The remaining sections are as follows: Section 2 and Section 2.1 summarize relevant literature, data collection, and process to enlighten the study. Section 2.2 describes selected observed characteristics and variables for Section 2.3 (Statistical Analysis). Results in Section 3 connect Discussion (Section 4) to Conclusions (Section 5).

## 2. Materials and Methods

### 2.1. Literature and Study Data Collection

Literature from DHS inspired the study and answered the methodological question with statistical matching. A few previous studies made a (quasi) longitudinal study possible after cross-sectional data aggregation from recent trends, descriptive, and intervention studies were included. When researchers cannot gather information from the identical participants sequentially, the alternative is to find and match a cohort in approximate conditions [27]. A study used statistical matching to identify Senegalese women’s health and family planning autonomy in the context of a male-dominated country. The 2017 Senegal DHS demonstrated that only 6% of women were independent; while most others were dependent on their husbands/partners, the matching results indicated family planning was more prevalent among autonomous women [28]. Another study revealed a strong impact of television and radio communications on family planning for Senegalese women in the 2014 DHS. Contextually, the real value is to realize empowerment of communications on Senegalese, Ghanaian, and Liberian females toward family planning [29].

The study considered multiple years of Liberia (2007, 2013), Ghana (2008, 2014), and Senegal DHS (2010–11, 2016). The six separate datasets (individual records of 15- to 49-year-old women) were downloaded and cross-checked for variables and sample size. There was exhaustive data checking and cleaning for outliers and missing information. Afterward, more than 5000 variables per dataset were truncated into 13 variables. In addition to rigorous correlation and simple regression testing between the chosen dependent variable and other variables, the literature review helped determine the study variables. The six separate datasets were merged into one file that was then separated into year 0 (2007 Liberia, 2008 Ghana, and 2010–11 Senegal) and year 1 (2013 Liberia, 2014 Ghana, and 2016 Senegal). The one merged file and information in year 0 were considered for any significant changes before the primary analysis for year 1.

Merging datasets required stepwise procedures. First, CASEID, the original variable per dataset, was combined with cluster, household number, and respondent’s line number. The information was recorded as a 15-character string variable. Two more pieces of information, either year 0 or 1 and country code (Liberia: 1, Ghana: 2, Senegal: 3), were added in front of the existing identifier before conversion to numeric. After conversion, this new CASEID became a unique case identifier as the basis for the data merger. Using Stata 17, *merge m:1 varlist…* command made a new CASEID corresponding to a multitude of observations in the master dataset. Another Stata command, *duplicates*, also made it possible to check any duplicated observations in the merged dataset. While it might be possible for the same respondents in each year, they were regarded as unique observations.

### 2.2. Covariates and Variables

The question remains on how public programs invest and intervene in family planning. The first step is to identify knowledge, perception, and practice domains. With having them as independent variables, an additional task follows to find all possible covariates—those representing control and treatment groups. The goal of covariate matching is to determine the effectiveness of the treatment: radio communications in family planning.

The aforementioned information from the 2008 Ghana DHS decided the dependent variable: the optimal number of children. The DHS questionnaire asked 11,594 married women as two groups (living with children and without children) to choose the exact number of children in their entire lives if they were to rewind the clock. The dependent variable—the ideal number of children—was derived from two questions that were asked based on whether women had children or not. Those who had children were asked to identify the ideal number retrospectively, while those who did not have children were asked to identify the ideal number prospectively. Three DHS were considered and analyzed (Liberia 2013; Ghana 2014; Senegal 2016) in one merged dataset (personal identification numbers as the basis for data aggregation). Earlier collections (Liberia 2007; Ghana 2008; Senegal 2010–11) were to ensure any significant trends between two periods. Note: two periods (Liberia 2007; Ghana 2008; Senegal 2010–11) and (Liberia 2013; Ghana 2014; Senegal 2016) are specified as *earlier* and *later*, hereafter. Hypothetically, radio communications change the perception toward the number of wanted children in the similar survey collection period. Other than this communication variable, more explanatory variables that could independently influence the optimal number were the number of deceased children, contraceptive use, and desire for family planning. More hypotheses exist between the dependent and independent variables because those who lost children may want more children while other independent variables could cause a reduction in the number of wanted children. In the result, plus and minus operations helped confirm or disapprove those hypotheses and the effectiveness of radio communications together with other explanatory variables.

Covariates are observable characteristics to match control and treatment groups statistically. The DHS literature advised the selection of covariates, and intimate partner violence (IPV) appeared at the forefront. There were five IPV reasons: (1) wives went out without the permission of husbands/partners, (2) neglected children, (3) arguments, (4) refused sex, and (5) burnt food. The covariate IPV informs the accumulation of those cases in one. Literacy was complementary and had a negative statistical relationship with IPV. There were disproportionate IPV cases among those who could read a partial or whole sentence (2463/3730 literates, 66% answered no IPV, as opposed to 3775/7861 illiterates, 48%). In all five IPV cases, 1097 illiterates (14%) were compared with 172 (5%) literates. Residence linked additional characteristics. A total of 2825 women in the capital had higher literacy but fewer IPV in addition to more education and individual wealth than 8769 non-capital residents. Additional characteristics included fewer children and balancing religion and family decisions— namely, residence information further completed the community, physical, social, and environmental milieu [30]. The last two covariates comprised the couple’s ages, respectively. Table 1 and Table 2 show summary statistics of dependent and independent variables, as well as covariates.

Earlier DHS releases (Liberia 2007; Ghana 2008; Senegal 2010–11) showed insignificant trend changes compared with later years. Among a total 11,014 respondents, 25% (2756) resided in the capital and 75% (8258) outside of the capital. The optimal number of children had about the same minimum (=0), mean (=5), and standard deviation (=2.4) despite different maximum (=25) among 9560 respondents, nor were there great differences in age, partner’s age, and child deaths between the two periods. Literacy and IPV continued negative correlation. More than 50% literates experienced no IPV (vs. 34% illiterates) but more than 17% illiterates had all five cases (vs. 7% literates). There were 3201 literates and 7813 illiterates.

There were three remarkable changes between earlier and later periods. More contraception is one. In earlier years, merely 17% used contraception (1925/11,014 respondents). However, as shown in Table 2, the rate was 10% more. Another change was the desire for more or no additional children. Between both periods, the desire for no more children gained 3% more than the other side (3284 no more children or 30% vs. 7730 more children wanted or 70% in earlier years). The increase in radio communications was impressive. In only six years, Ghana, Liberia, and Senegal had 15% more females who received family planning communication mainly by radio. The change reversed the magnitude of control and treatment groups (6857 non-received or 62% vs. 4157 received or 38% in the past). Overall, every comparison reinforced the setting and advancement of this study.

### 2.3. Statistical Analysis

Exact matching searches and organizes people whose observable characteristics/covariates look identical. The validity of matching should result in no discernible differences between control and treatment groups that could correlate with the outcome and unobservable factors (excluding participation) [31,32].

Coarsened exact matching (CEM) is an exact matching method for limiting the maximum monotonic imbalance using covariates to enhance causal mechanisms. CEM depends on the user’s ex ante covariate selection to coarsen the empirical distributions. The more fitting a group of ex ante analysis covariates (as the bins become tighter), the more aligned the maximum imbalance ex post. The matching solution offers the whole multivariate histogram, accommodating continuous and categorical covariates. Ideally, the user can expect balances between two groups in the identical conditions of covariates [33,34,35]. 

Consider the CEM algorithm in detail. Samples of *n* are randomly gathered from the target population *N* (*n* must be equal to or less than *N*). $T_{i}$ = 1 indicates a sample/an individual who received treatment; $T_{i}$ = 0 otherwise. Both treatment and control groups can remain after the accumulation of all samples. The respective outcome can be $Y_{i}\left(1\right)$ for those who received treatment and $Y_{i}\left(0\right)$ for non-treated. The treatment effect for individual i or $TE_{i}$ is the difference between $Y_{i}\left(1\right)$ and $Y_{i}\left(0\right)$, where the observed result is $Y_{i}=T_{i}Y_{i}\left(1\right)+\left(1-T_{i}\right)Y_{i}\left(0\right)$. Note: $Y_{i}\left(0\right)$ and $Y_{i}\left(1\right)$ are unobserved whether individual i receive treatment or not. Assume that control and treatment groups differ before treatment. An empirical dataset combines $X_{c}$, which is a column vector of covariate c with every sample n observation. That is, $X=\left[X_{ic},\;i=1,\ldots ,n,\;c=1,\ldots ,k\right]$. (Additionally, a k-dimensional dataset is expressed as $X=\left(X_{1},X_{2},X_{3},\ldots ,X_{k}\right)$.) Altogether the sample average treatment effect on the treated (SATT) appears to be where $n_{T}=\overset{n}{\underset{i=1}{\sum }}T_{i}\;and\;T=\left\{1\leq i\leq n:T_{i}=1\right\}$:(1)$$SATT=\frac{1}{n_{T}}\underset{i\in T}{\sum }TE_{i}$$

The treatment appointment and conceivable results are independent. That is,
(2)P(T|X, Y(0),Y(1))=P(T|X)

Various $L_{1}$ statistics as values of a multidimensional histogram indicate the perfect global balance $L_{1}=0$ and perfect imbalance $L_{1}=1$ in the treatment and control groups. The k-dimensional comparative frequencies count the absolute difference in the treated $f_{l_{1}\ldots l_{k}}$ and the control $g_{l_{1}\ldots l_{k}}$ samples. That is,
(3)$$L_{1}\left(f,g\right)=\frac{1}{2}\underset{l_{1}\ldots l_{k}}{\sum }\left|f_{l_{1}\ldots l_{k}}-g_{l_{1}\ldots l_{k}}\right|$$

Ideally, the matched frequencies for both samples $f^{m}$ and $g^{m}$ yield a lower $L_{1}$ statistic than the initial imbalance, which is:(4)$$L_{1}\left(f^{m},g^{m}\right)<L_{1}\left(f,g\right)$$

The final multivariate linear regression comes after the procedures above:(5)$$\;\;\;Wantnumchild_{i}=\beta_{0}+\beta_{1}trt+\beta_{2}currentuse+\beta_{3}childdeaths+\beta_{4}fpdesire+\varepsilon$$

## 3. Results

Data were analyzed with Stata 17. The results are divided into two subsections: (1) CEM and (2) multivariate linear regression. The original Stata outputs are in Appendix A with other essential information.

### 3.1. CEM Results

Table 3 details the multivariate and univariate $L_{1}$ distances to determine CEM operations. The multivariate $L_{1}$ (0.477) indicates neither perfect imbalance nor perfect balance. The baseline reference is in between the two extremes. The five unidimensional measures next to covariate names show the lowest (Specialist) to the highest (Ipvcases) imbalances. No high univariate $L_{1}$ was observed. A divergence appears for the experimental and treatment groups for the minimum, 25th, 50th, 75th, or 100th percentile.

The imbalance ex ante hinted at CEM operations. The automated CEM algorithm independent of human control arranged the observations into separate strata whose predetermined covariates looked the same and released others not in the strata. Stata returned CEM strata, CEM matched and unmatched cases, and CEM weights assigned to the stratum.

The automated CEM algorithm matched 4840 control and 5492 treated cases out of 11,594 instances (560 control and 702 treated cases unmatched). The number of matched strata was 875 from a total of 1674. In a standard setting, the algorithm uses the most considerable covariate information possible, leaving different numbers of cases and strata on both sides. CEM weights are complementary to different sizes.

Table 4 contains the automated CEM results. Compared with the initial imbalance in Table 3, the algorithm lowered the multivariate $L_{1}$ statistic by 0.116 and the univariate $L_{1}$ statistic for each covariate. No percentile difference between control and treated groups in each covariate remained. These pieces of evidence clarified the appropriateness of CEM operations to balance two groups in advance of testing treatment effects.

K-to-K matching equalizes the treated and control observations. This addition was to double-check whether CEM was suitable for reducing the imbalance between both groups. As a result, 3776 observations on both sides remained, respectively. The number of matched and total strata was the same as the automated CEM. Despite many more unmatched cases, this matching solution bore a similar multivariate $L_{1}$ distance (1624 control and 2418 treated observations were out of selection after K-to-K). The results in Table 4 and Table 5 indicate that the automated CEM should suffice to lead to an additional multivariate linear regression.

### 3.2. Multivariate Linear Regression Results

The linear regression results after CEM further help determine the effectiveness of radio communications on family planning. The simple/univariate linear regression between the dependent variable (i.e., the exact number of children that respondents/married women wanted) and the treatment variable indicates the correctness of the SATT estimate, which was −0.34. Instead of magnitude change, the interpretation focused more on plus and minus operations because of narrow margins and the small mean of the dependent variable. At first glance, radio listeners perceived fewer children wanted. Then the multivariate regression expanded this causality with other independent variables.

The SATT estimate in the multivariate regression was about the same as the univariate result. Contraception had a similar coefficient and t-statistic, meaning radio communications and current contraceptive use had the same effect on reducing the number of children wanted. The desire for no additional children reduced the number of children too. In contrast, mothers who lost children desired more children. All independent variables were statistically significant at a more than 99% confidence interval. Total observations were 9438.
(6)$$\begin{array}{cccccc} \hat{\mathrm{wantnumchild}}= & 5.4 & -\;0.34\;\mathrm{trt} & -\;0.35\;\mathrm{currentuse} & +\;0.58\;\mathrm{childdeaths} & -\;0.47\;\mathrm{fpdesire}\\ & \left(142.07\right) & \left(7.73\right) & \left(7.22\right) & \left(22.46\right) & \left(10.05\right) \end{array}$$
**Note:** T-statistics are below coefficients in parenthesis. All *p*-values < 0.001. F-statistic (4, 9473): 168.66. All standard errors are between 0.03 and 0.05 (rounded). n = 9438.

## 4. Discussion

This study corroborated previous findings in the literature that family planning communications could assist a couple’s family planning decisions. This study investigated further three countries whose social status among females differs. Liberia notably ran a nationwide family-planning campaign supported by the government and the United States Agency for International Development. The *Baby by Choice, Not by Chance* slogan and *Family Planning is Good for Baby Ma!* educational posters informed more women on accessing family planning education and practice. The DHS information provided the possibility of radio as a medium of family planning information to reach more target populations toward more met needs among families. Considering the context that more homemakers, female farmers, and others working inside and outside houses are illiterate, communicate in dialects, and are familiar with indigenous culture, local radio stations can couple with other information and communications technologies (ICTs) to listen to listeners’ concerns and interests, learn about up-to-date news, filter unnecessary and inaccurate information, and communicate more closely with stakeholders. 

The overall study experienced a complete step-by-step process. CEM algorithms balanced the imbalance between the treated (listeners) and control (non-listeners). Beforehand, the literature guided the selection of covariates. Those observed characteristics were suitable for advancing regression analysis—the simple linear regression supports the association that radio is an effective tool in a full-coverage public health campaign to educate sub-Saharan African women. Then the multivariate linear regression offered another opportunity to test the treatment effect with other independent variables. Most notably, communications and contraception influenced the dependent variable by about the same amount. Further realistically, the desire for no additional children decreased wanted children; the opposite was reasonable for those who lost children. Each linear regression referred to most of the matched observations. (Losses happened due to missing information among variables.) Being nationally representative, the DHS provided sufficient information to run the exact matching method.

Limitations exist for future study. First, DHS has repeated cross sections. Participants likely differ over time even though the same questions are in the questionnaire. The limit calls for another study to have the same participant complete the survey over time to observe the effects of family planning communications. However, this study’s questions depended on past experiences to the survey time. Second, more information should be included on how couples make family planning decisions. Various couples’ ages, collaborative decision-making processes, different social statuses, and more will further enlighten family planning by 2100. Last but not least, this study found an appropriate dependent variable to enlighten the results of the study. However, there could be a better-suited dependent variable for future study.

National and community-based public health campaigns should consider culture; primary, secondary, and tertiary occupations and education; and geographic specificity. For instance, DHS surveyors and researchers should evaluate how target populations with every aspect receive the same public messages differently. Building trust in the target areas is indispensable to public health interventions [36,37,38,39,40,41].

## 5. Conclusions

This study concludes that knowledge and perception domains are equally important to practicing contraception. Policymakers and stakeholders should consider appropriate policies and support for couples to make harmonized family planning decisions. Radio has diffusion and dissemination power in sub-Saharan Africa. Thus, similar or better delivery tools are recommended to communicate family-planning information to reach more target populations. This study does not overemphasize family planning as the best option for sustainable growth. But this study considered family planning in stagnant food production and continued population growth, balanced work, life, and education.

## Figures and Tables

**Table 1 ijerph-19-04577-t001:** Integer/count covariates and variable of interest (Liberia 2013; Ghana 2014; Senegal 2016).

Name	Role	Min	Mean	Max	Std Dev
agen	Covariate	15	31	49	8
partage	Covariate	15	43	99	17
ipvcases	Covariate	0	1.5	5	1.9
childdeaths	Independent	0	0.43	10	0.89
wantnumchild	Dependent	0	5	30	2.21

**Note: agen** (n = 11,594): age; **partage** (n = 11,579): spouses’ age; **ipvcases** (n = 11,593): intimate partner violence cases; **childdeaths** (n = 11,594): sons and daughters died; **wantnumchild** (n = 10,486): exact (optimal/ideal) number of children that respondents wanted.

**Table 2 ijerph-19-04577-t002:** Binary covariates and variables of interest (Liberia 2013; Ghana 2014; Senegal 2016).

Name	Role	No		Yes		Full Cases	
		n	%	n	%	n	%
specialdist	Covariate	8769	76	2825	24	11,594	100
literacy	Covariate	7862	68	3730	32	11,592	100
trt	Treatment	5400	47	6194	53	11,594	100
currentuse	Independent	8481	73	3113	27	11,594	100
fpdesire	Independent	7823	67	3769	33	11,592	100

**Note: specialdist:** (no) residents outside capital/(yes) residents in the capital; **literacy:** (no) women cannot read/(yes) women can read a part or whole sentence; **trt:** (no) respondents received no family planning information/(yes) respondents received family planning information via radio communications primarily and multiple exposures through television, newspaper, health facilities, and health workers; **currentuse:** (no) no contraception/(yes) current contraception; **fpdesire:** (no) no more children wanted/(yes) want more children.

**Table 3 ijerph-19-04577-t003:** Initial imbalance in covariates between treatment and control groups.

$\mathrm{Multivariate}\;L_{1}\;\mathrm{Statistic}:\;0.477$							
Covariate	$\mathrm{Univariate}\;L_{1}\;\mathrm{Distance}$	Mean Difference	Min	25%	50%	75%	Max
agen	0.08	0.98	1	1	1	1	0
partage	0.09	−4	0	0	−1	−4	0
ipvcases	0.12	−0.45	0	0	0	−1	0
specialdist	0.06	0.06	0	0	0	1	0
literacy	0.14	0.14	0	0	0	1	0

**Table 4 ijerph-19-04577-t004:** Automated CEM summary.

$\mathrm{Multivariate}\;L_{1}\;\mathrm{Statistic}:\;0.361$							
Covariate	$\mathrm{Univariate}\;L_{1}\;\mathrm{Distance}$	Mean	Min	25%	50%	75%	Max
agen	0.04	0.05	0	0	0	0	0
partage	0.03	−0.06	0	0	0	0	.
ipvcases	<0.001	<0.001	0	0	0	0	0
specialdist	<0.001	<0.001	0	0	0	0	0
literacy	<0.001	<0.001	0	0	0	0	0

**Table 5 ijerph-19-04577-t005:** K-to-K matching summary.

$\mathrm{Multivariate}\;L_{1}\;\mathrm{Statistic}:\;0.324$							
Covariate	$\mathrm{Univariate}\;L_{1}\;\mathrm{Distance}$	Mean	Min	25%	50%	75%	Max
agen	0.04	0.07	0	0	1	0	0
partage	0.02	0.04	0	0	0	0	.
ipvcases	0	0	0	0	0	0	0
specialdist	0	0	0	0	0	0	0
literacy	0	0	0	0	0	0	0

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
