# Peer review of "Radio Communications on Family Planning: Case of West Africa"

_ijerph, 2022, doi:10.3390/ijerph19084577_

Round 1
Reviewer 1 Report
The contribution addresses a very important theme, which perhaps requires a specific study of the idea of the family (why authors choose to interview only married couples?), of female self-determination with respect to motherhood, national programs as Baby by Choice, Not by Chance and Family Planning is Good for Baby Ma!
- line 88 Literature from DHS that inspired the study
- line 101 National campaings (for exemple: why they address the attention only on married couples?)
- line 344 In my (pedagogical) opinion, family planning is not only a matter of food availability or polulation growth, but also a matter of life style, educational care, female work etc...
Author Response
Comment:
The contribution addresses a very important theme, which perhaps requires a specific study of the idea of the family (why authors choose to interview only married couples?), of female self-determination with respect to motherhood, national programs as Baby by Choice, Not by Chance and Family Planning is Good for Baby Ma!
Response:
We much appreciate your valuable comment. Your comment helped us to specify the target population and correct information. Please bear our excuse of line changes due to track changes.
Comment:
- line 88 Literature from DHS that inspired the study
- line 101 National campaings (for exemple: why they address the attention only on married couples?)
Response:
We reflected two lines in a new sentence: "This study's target populations were the same as the national campaigns, 15-49-year-old women who are married or cohabitating with partners. Simultaneously, they are the ones who make family planning decisions in support of and with their husbands/partners." in lines 63-66 on page 2.
Comment:
"line 344 In my (pedagogical) opinion, family planning is not only a matter of food availability or polulation growth, but also a matter of life style, educational care, female work etc...
Response:
In lines 381-383 on page 9, we revised the sentence, "This study does not overemphasize family planning as the best option for sustainable growth but expectations for stagnant food production and continued population growth, balanced work, life, and education."
Reviewer 2 Report
This is a study of an immensely important topic - how to facilitate population control in view of the current and future sustainability conditions. The researchers have identified 3 different locations in the SSA areas which represent different educational and gender relations background to attempt to see how contraception decisions are made, especially on the influence of radio communication to the results of such control. The comparison of two sets of data regarding these 3 locations offers a good proof that indeed there is a positive correlation between the transmission of family planning concepts via radio communication and the results of number of children, over the years. It can be seen that the paper demonstrate clear connection between the two, and therefore lends good support to improving such mass communication mechanisms such as radio communication, to target different demographic groups, for enhancement of family planning effects. The overall presentation of the paper is clear, with good organisations of the data sets. The conclusion is particularly useful as it points out other details that should be included in such studies down the road. I think this is a very good preparation for further, large scale research on this topic, especially seeing the significance of this issue.
Author Response
Thank you very much for your insightful comment on this study. Based on almost all my research career in Ghana, Liberia, and Senegal, I experienced the power of radio communication in the field and decided on applications. I have heard about and talked about sustainable development, and this study gives us proof of “how-to” do so.
I confirm two additional values, according to your insight. First, this study could incorporate three different contextual countries but present coherent results and conclusions. A coarsened exact matching matched well among various covariates for multiple linear regression. Second, the overall study design, results, and conclusions with national representative Demographic and Health Surveys Data evince extended nationwide family planning campaigns.
Thank you again so much for your insight. Your comment provides us a moment to affirm the values of this study.
Reviewer 3 Report
This secondary data analysis examined an important issue. Although I highly confirmed the aims and efforts of this study, the current manuscript is hard to followed.
- I would like to suggest the authors revise the contents of Materials and Methods. Most of contents in 2.1. Data and literature should be moved to Introduction section to make the readers understand the background and data sources of this study.
- Because the data were composed of various sources collected in various stages, a table describing the details are needed.
- The aims of this study should be clearly described in Introduction section.
- the results of statistical analysis should be clearly described in the words the readers can understand.
- “In 2020, many 15-49-year-old women of reproductive age and their partners delayed or avoided pregnancy.” “More use of modern contraception ensued in the northern hemisphere.” These two sentences seemed com form reference 5. Please label them by reference 5.
Author Response
(Comment)
This secondary data analysis examined an important issue. Although I highly confirmed the aims and efforts of this study, the current manuscript is hard to followed.
Response:
Thank you very much for your confirmation of the importance of this study. Co-authors and I handled your five points in revision with care. We appreciate every suggestion you made in this manuscript better and more detailed.
(Point 1)
I would like to suggest the authors revise the contents of Materials and Methods. Most of contents in 2.1. Data and literature should be moved to Introduction section to make the readers understand the background and data sources of this study.
Response:
We moved most of the information in 2.1. Data and literature to 1. Introduction. We also added the details of the target population to that section. New sentences are in red, so you can easily notice the change. We reflected this request in lines 82-116 on pages 1-2.
(Point 2)
Because the data were composed of various sources collected in various stages, a table describing the details are needed.
Response:
We clarified the sentence that “Demographic and Health Surveys (DHS) from USAID were the study’s primary data that have represented invaluable household information over 90 low- and middle-income countries middle-income countries worldwide since 1984.” in lines 82-84 on page 2. This sentence clearly states that USAID provided publicly available DHS data. Also, we added new sentences about how we developed data for analysis in lines 138-162 on pages 3-4.
(Point 3)
The aims of this study should be clearly described in Introduction section.
Response:
In lines 77-81 on page 2, we added a new sentence, “The aim of increasing public awareness of family planning is to investigate more realistically and compare the effectiveness of radio on contraception, loss of children, and the desire for family planning. If the results and interpretations suffice, this study could be evidence for nationwide family planning campaigns reinforcing radio communication.”
(Point 4)
the results of statistical analysis should be clearly described in the words the readers can understand.
Response:
In Sections 3.1. CEM Results and 3.2. Multivariate Linear Regression Results, we made track changes to describe the results more clearly. We made significant grammar corrections and word changes.
(Point 5)
“In 2020, many 15-49-year-old women of reproductive age and their partners delayed or avoided pregnancy.” “More use of modern contraception ensued in the northern hemisphere.” These two sentences seemed com form reference 5. Please label them by reference 5.
Response:
We labeled them correctly in lines 46 and 47 on page 1.
Round 2
Reviewer 3 Report
The authors have revised their manuscript based on the reviewer's suggestions. I would like to suggest the editors accepting it for publication.